# Physical origin of giant excitonic and magneto-optical responses in two-dimensional ferromagnetic insulators

Meng Wu [1,2], Zhenglu Li [1,2], Ting Cao [1,2] & Steven G. Louie [1,2]

The recent discovery of magnetism in atomically thin layers of van der Waals crystals has created great opportunities for exploring light–matter interactions and magneto-optical phenomena in the two-dimensional limit. Optical and magneto-optical experiments have provided insights into these topics, revealing strong magnetic circular dichroism and giant Kerr signals in atomically thin ferromagnetic insulators. However, the nature of the giant magneto-optical responses and their microscopic mechanism remain unclear. Here, by performing first-principles $GW$ and Bethe-Salpeter equation calculations, we show that excitonic effects dominate the optical and magneto-optical responses in the prototypical two-dimensional ferromagnetic insulator, $CrI_3$. We simulate the Kerr and Faraday effects in realistic experimental setups, and based on which we predict the sensitive frequency- and substrate-dependence of magneto-optical responses. These findings provide physical understanding of the phenomena as well as potential design principles for engineering magneto-optical and optoelectronic devices using two-dimensional magnets.

[1] Department of Physics, University of California at Berkeley, Berkeley, CA 94720, USA. [2] Materials Sciences Division, Lawrence Berkeley National Laboratory, Berkeley, CA 94720, USA. Correspondence and requests for materials should be addressed to S.G.L. (email: sglouie@berkeley.edu)

The magneto-optical (MO) effects, such as the magneto-optical Kerr effect (MOKE) and the Faraday effect (FE), have been intensively investigated experimentally in a variety of magnetic materials, serving as a highly sensitive probe for electronic and magnetic properties. Recent measurements using MOKE have led to the discovery of two-dimensional (2D) magnets, and demonstrated their rich magnetic behaviors[1,2]. In particular, a giant Kerr response has been measured in ferromagnetic mono- and few-layer $CrI_3$[2]. Magnetic circular dichroism (MCD) in photo absorption has also been measured in ferromagnetic monolayer $CrI_3$[3]. However, the exact microscopic origin of such large MO signals and MCD responses in 2D materials is still unclear, because treating accurately sizable spin−orbit coupling (SOC) and excitonic effects that are essential for such an understanding in these systems has been beyond the capability of existing theoretical methods.

$CrI_3$, in its monolayer and few-layer form, is a prototypical 2D ferromagnetic insulator with an Ising-like magnetic behavior and a Curie temperature of about 45 K, exhibiting tremendous out-of-plane magnetic anisotropy[2]. Within one layer, the chromium atoms form a honeycomb structure, with each chromium atom surrounded by six iodine atoms arranged in an octahedron (Fig. 1a, b), and the point group of the structure is $S_6$. The crystal field therefore splits the Cr 3d and I 5p ligand states into $t_{2g}$ and $e_g$ manifolds; the spin degeneracy of which are further lifted by the exchange interaction. Although the major-spin $e_g$ states are delocalized due to strong $p−d$ hybridization, the magnetic moment is approximately $3\mu_B$ at each Cr site, in accordance with an atomic picture from the first Hund's rule[4].

With our recently developed full-spinor $GW$ and Bethe-Salpeter equation (BSE) methods, we show from first principles that the exceedingly large optical and MO responses in ferromagnetic monolayer $CrI_3$ arise per se from strongly bound exciton states consisting of spin-polarized electron−hole pairs that extend over several atoms. These exciton states are shown to have distinct characteristics compared with either the Frenkel excitons in ionic crystals and polymers, or Wannier excitons in other 2D semiconductors. By simulating realistic experimental setups, we further find that substrate configuration and excitation frequency of the photon strongly shape the MO signals. Our results provide the conceptual mechanism for the giant optical and MO responses, explaining quantitatively the recent experiments on $CrI_3$[2,3]. In addition, comparison between bulk and monolayer $CrI_3$ reveals the pivotal role of quantum confinement in enhancing the MO signals.

## Results

**Quasiparticle band structure**. An accurate first-principles calculation of the electronic structure of $CrI_3$ should account for both the dielectric polarization from the ligand groups and the on-site Coulomb interactions among the localized spin-polarized electrons. We adopt the following approach. The first-principles $GW$ method has become a de facto state-of-the-art approach to describe dielectric screening and quasiparticle excitations in many real materials[5]. In practical calculations here, the $G_0W_0$ approximation[5] is employed where the $GW$ self-energy is treated as a first-order correction, and the single-particle Green's function $G$ as well as the screened Coulomb interaction $W$ are calculated using eigenvalues and eigenfunctions from density-functional theory (DFT). Through the screened Coulomb interaction $W$, the nonlocal and dynamical screening effects as well as the self-energy effects beyond the DFT Kohn−Sham orbital energies (within the local-spin-density approximation (LSDA)) are captured. Also, in previous studies, the method of LSDA with an on-site Hubbard potential (LSDA + $U$) has served as a reasonable mean-field starting point for $G_0W_0$ calculations in correlated systems to avoid the spurious $p−d$ hybridization[6,7]. In this work, we adopt an on-site Hubbard potential in the rotationally invariant formulation[8] with $U = 1.5$ eV and $J = 0.5$ eV, with fully relativistic pseudopotentials and a plane wave basis set. The validity of this specific set of $U$ and $J$ has been carefully tested (see Supplementary Figs. 1 and 2). Throughout this work, the magnetization of ferromagnetic monolayer $CrI_3$ is taken to be along the $+z$ direction (Fig. 1b). As shown in Fig. 1c, our calculations reveal a strong self-energy correction to the quasiparticle bandgaps, due to the weak dielectric screening in reduced dimensions and the localized nature of the d states. The direct bandgap is 0.82 eV at the Γ point at the LSDA + $U$ level, whereas the direct $G_0W_0$ quasiparticle bandgap including the self-energy effect is 2.59 eV, as shown in Fig. 1c. Throughout the calculations, we incorporate the SOC effect from the outset by employing full two-component spinor wave functions.

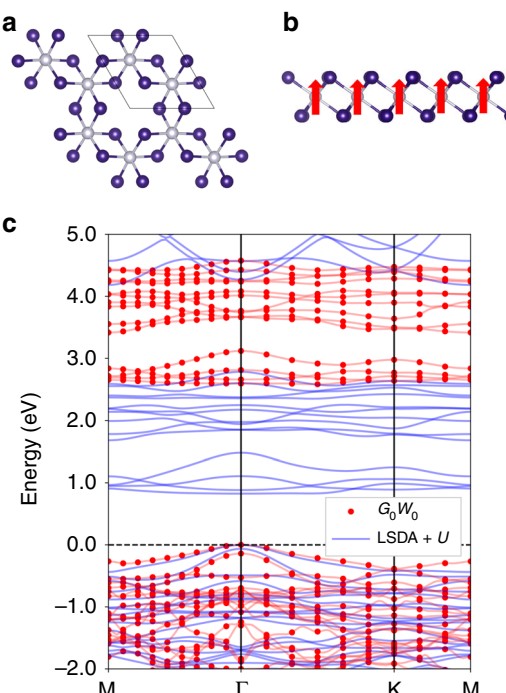

**Fig. 1** Crystal structure and electronic structure of ferromagnetic monolayer $CrI_3$. **a** Crystal structure (top view) of monolayer $CrI_3$. Chromium atoms are in gray while iodine atoms in purple. **b** Crystal structure (side view) of ferromagnetic monolayer $CrI_3$. Red arrows denote the out-of-plane magnetization, which is pointing along the $+z$ direction. **c** $G_0W_0$ (red dots) and LSDA + $U$ (blue lines) band structures of ferromagnetic monolayer $CrI_3$. A rotationally invariant Hubbard potential is employed with $U = 1.5$ eV and $J = 0.5$ eV in the LSDA + $U$ calculation, which is then used as the starting mean field for the $G_0W_0$ calculation. The $G_0W_0$ band structure is interpolated with spinor Wannier functions

**Exciton-dominant optical responses**. The strong SOC strength and the ligand states strongly hybridizing with Cr d orbitals (see Supplementary Fig. 3) have decisive influences on the electronic structure and optical responses of ferromagnetic monolayer $CrI_3$. SOC significantly modifies the bandgap and band dispersion near the valence band maximum[4]. Figure 2a shows the $G_0W_0$ band structure together with each state's degree of spin polarization (with an out-of-plane quantization axis), of which the orbital and spin degeneracy are consistent with the above discussions. After solving the first-principles BSE, which describes the electron−hole interaction[9], with spinor wave functions, we find a series of

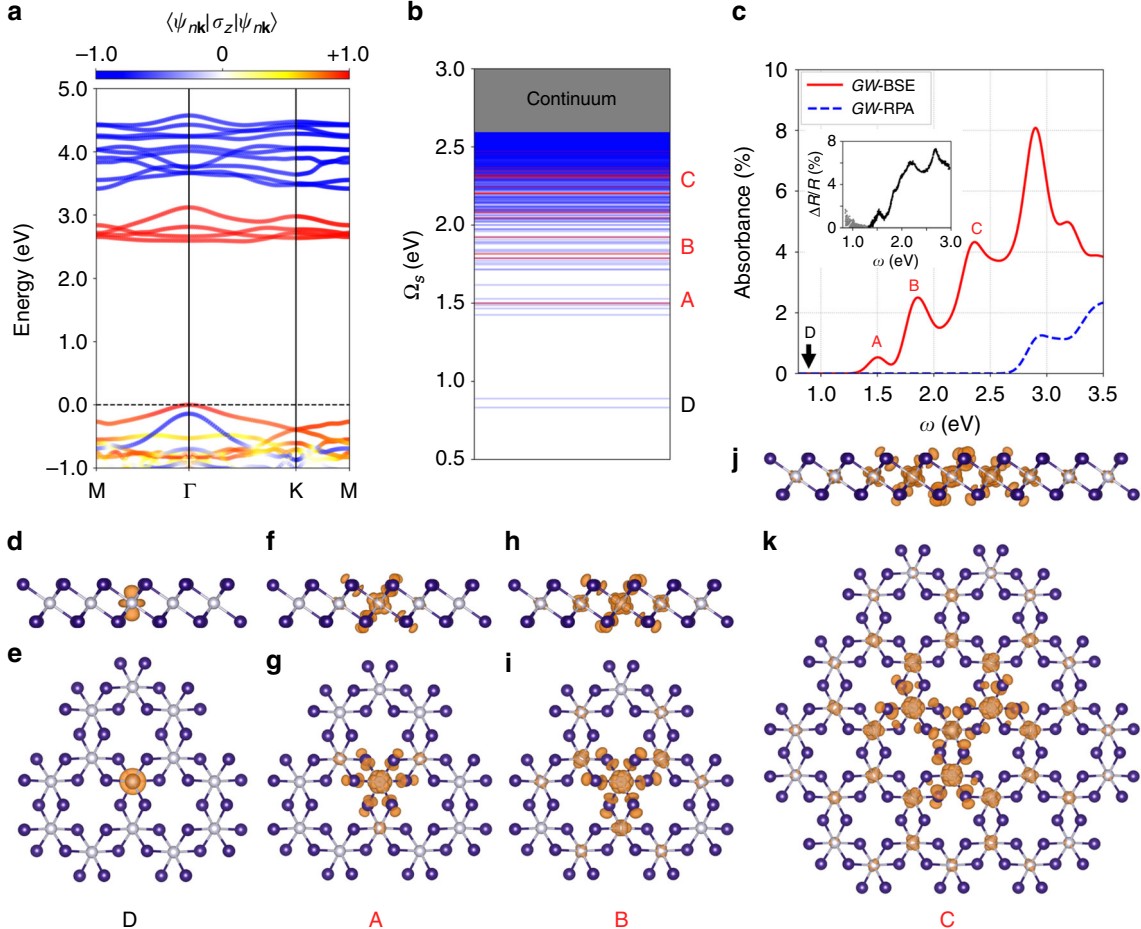

**Fig. 2** Calculated quasiparticle band structure and excitons in ferromagnetic monolayer CrI₃. **a** $G_0W_0$ band structure with colors denoting the magnitude of spin polarization along the out-of-plane direction. The red (blue) color denotes the major-spin (minor-spin) polarization. **b** Exciton energy levels of monolayer CrI₃ calculated using the first-principles GW-BSE method. Optically bright exciton states are in red while dark ones in blue. The bright excitons have at least two orders of magnitude stronger oscillator strength compared with the dark ones. The free-electron−hole continuum starts from 2.59 eV. We label the bound exciton states with D for the lowest-lying dark states and A−C for the higher-lying bright states as evident in the plot of exciton levels. **c** Absorption spectrum of linearly polarized light with electron−hole interaction (GW-BSE, solid red line) and without electron−hole interaction (GW-RPA, dashed blue line). The inset data are extracted from ref. [3] showing the experimental differential reflectivity measured on a sapphire substrate, and the signals above 1.3 eV are shown in black for better comparison. **d–k** Exciton amplitudes in real space with the hole fixed on a Cr atom. Shown are iso-value surfaces of the amplitude square with the value set at 1% of the maximum value. Upper panel: side view. Lower panel: top view. **d, e** Dark exciton D with an excitation energy $\Omega_S$ at 0.89 eV; **f, g** bright exciton A with $\Omega_S$ at 1.50 eV; **h, i** bright exciton B with $\Omega_S$ at 1.82 eV; **j, k** bright exciton C with $\Omega_S$ at 2.31 eV. Here the dominant states (with the largest oscillator strength among the nearby states in the same group) are plotted

strongly bound dark (optically inactive) and bright (optically active) exciton states with excitation energies ($\Omega_S$) below the quasiparticle bandgap, as shown in the plot of the exciton energy levels (Fig. 2b). As seen in Fig. 2c, the calculated linearly polarized absorption spectrum including electron−hole interactions (i.e., with excitonic effects, solid red curve labeled GW-BSE) features three peaks at around 1.50, 1.85 and 2.35 eV (below the quasi-particle gap of 2.59 eV), which are composed of several bright exciton states in each peak and denoted as A, B and C, respectively. This is in contrast to the calculated step-function-like noninteracting absorption spectrum (i.e., without excitonic effects, dashed blue curve labeled GW-RPA). The magnitude of the absorbance peak around 1.50 eV is deduced to be 0.7% from a previous differential reflectivity measurement (Fig. 2c, inset)[3], while our calculated absorbance with a broadening factor of 80 meV is around 0.6% at 1.50 eV. From our calculation (Fig. 2b), there are also two dark states (excitons D) with enormous binding energy of larger than 1.7 eV. The existence of two states of nearly the same energy comes from the fact that there are two Cr atoms

in a unit cell. We plot the real-space exciton wave functions of these states, with the hole fixed on a Cr atom, in Fig. 2d–k. Unlike monolayer transition metal dichalcogenides where the bound excitons are of Wannier type with a diameter of several nanometers[10,11], ferromagnetic monolayer CrI₃ hosts dark Frenkel-like excitons localized on a single Cr atom (Fig. 2d, e) and bright charge-transfer or Wannier excitons with wave functions extending over one to several primitive cells (Fig. 2f–k). These plots are consistent with the intuition that a smaller exciton binding energy is related to a larger exciton radius[11,12]. Numer-ical calculations of the exciton radius further corroborate this conclusion (see Supplementary Table 1).

In addition, ferromagnetism and broken time-reversal sym-metry (TRS) play vital roles in determining the internal structure of the exciton states in ferromagnetic monolayer CrI₃, in contrast to the Frenkel/charge-transfer excitons determined solely by flat-band transitions in boron nitride systems[13,14], organic materials[15,16] or alkali halides[9]. The eigenstate of an exciton is a coherent superposition of free electron−hole pairs at different $k$

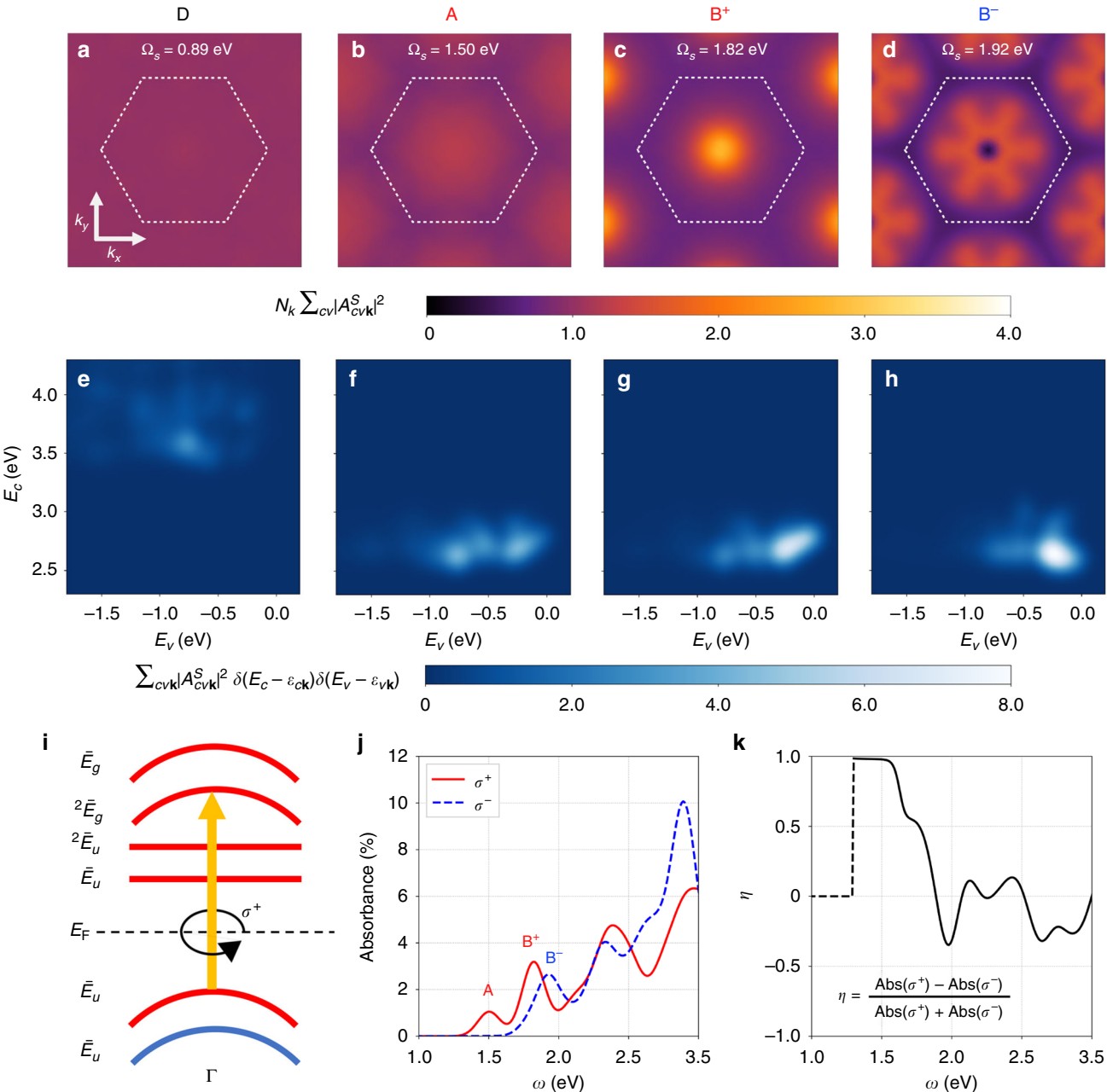

**Fig. 3** Internal structure of exciton states and MCD. **a–d** Exciton envelope functions in k-space of **a** exciton D with $\Omega_S = 0.89$ eV, **b** exciton A with $\Omega_S =$ 1.50 eV, **c** exciton $B^+$ with $\Omega_S = 1.82$ eV and **d** exciton $B^-$ with $\Omega_S = 1.92$ eV. The white dotted-line hexagon denotes the first Brillouin zone (BZ). The amplitudes are summed over band-pairs as given by $N_k \sum_{cv} \left| A_{cvk}^S \right|^2$, where $A_{cvk}^S$ describes the k-space exciton envelope function for the exciton state $|S\rangle$ and $N_k$ is the number of k-points in the first BZ. **e–h** The distribution of free electron−hole pair with electron energy at $E_c$ and hole energy at $E_v$ for selected exciton states: **e** exciton D, **f** exciton A, **g** exciton $B^+$ and **h** exciton $B^-$, weighted by module squared exciton envelope function for each interband transition between states $|vk\rangle$ and $|ck\rangle$, with quasiparticle energies $\varepsilon_{vk}$ and $\varepsilon_{ck}$, respectively. All the band energies are measured with respect to the valence band maximum. A bivariate Gaussian energy broadening with equal standard deviation of 80 meV is used to smoothen the distribution. **i** Schematics of interband transitions around the Γ point. The irreducible representations for Bloch states at the Γ point are labeled. $E_F$ and the dashed line denote the Fermi level. The color scheme for spin polarization is the same as in Fig. 2a. Among all possible transitions in (**i**), only the indicated $\sigma^+$ circularly polarized dipole transition is allowed. **j** Frequency-dependent circularly polarized absorbance of ferromagnetic monolayer $CrI_3$ at normal incidence. The solid red (dashed blue) curve corresponds to the $\sigma^+(\sigma^-)$ circularly polarized light. **k** MCD of photo absorbance ($\eta$) as a function of the photon frequency. $\eta$ is set to zero below 1.3 eV as shown by the dashed line

points ($|cv, \mathbf{k}\rangle$), and may be written as $|S\rangle = \sum_{cv\mathbf{k}} A_{cv\mathbf{k}}^S |cv, \mathbf{k}\rangle$, where $A_{cv\mathbf{k}}^S$ is the exciton envelope function in k-space[9]. Here $c$ denotes conduction (electron) states and $v$ denotes valence (hole) states. In Fig. 3a–d we plot the module square of the exciton envelope function in k-space. As expected of highly localized Frenkel excitons in real space, the lowest-lying dark state D in

Fig. 3a shows a uniform envelope function in k-space, whereas the bright states A ($\Omega_S = 1.50$ eV) and $B^+$ (at $\Omega_S = 1.82$ eV) in Fig. 3b, c have the envelope function localized around Γ and have $s$ characters. From Fig. 3d, an interesting hexagonal petal pattern with a node at Γ can be found for exciton $B^-$ ($\Omega_S = 1.92$ eV). In Fig. 3e–h, we plot the distribution of the constituent free electron

−hole pairs specified by ($E_v$, $E_c$) for selected exciton states, weighted by the module squared exciton envelope function for each specific interband transition. It is obvious that the electron −hole composition of exciton D is distinct from those of the bright states (A and B).

Because of broken TRS and strong SOC effect[9], the electron (hole) states that compose a given exciton in this system are from Bloch wave functions with spin polarization along different directions, giving rise to rich excitonic spin configurations. In fact, the lowest-lying bound exciton states are all formed by Kohn−Sham orbitals with particular spin-polarization. Our calculations verify that the dark excitons D are dominated by (>99.5%) transitions between the major-spin valence bands and minor-spin conduction bands. The bright states (forming peaks A, B and C) in Figs. 2f–k and 3b–d, f–h, however, are all dominated by (>96%) transitions between the major-spin valence bands and major-spin conduction bands (see Supplementary Table 2). Ligand field theory can provide a qualitative understanding of the lowest-lying D and A exciton states of which the optical transitions mainly occur among the localized Cr $d$ orbitals[3,17]. However, ligand field theory is insufficient to evaluate the oscillator strength of the excitons quantitatively. In addition, the coexistence of Frenkel and Wannier excitons in our system poses significant challenges to ligand field theory, while this excitonic physics can be fully captured by the first-principles $GW$-BSE method.

**MO effects from first principles.** The above-mentioned internal structures of the exciton states are essential for a deeper understanding of the MO responses. Note that all the irreducible representations of the double group $S_6^D$ are one-dimensional, which facilitates our analysis of optical selection rules for circularly polarized lights around the Γ point, as shown in Fig. 3i. For $1s$-like bright states A and B$^+$ wherein the transition mainly happens between the topmost valence band and the major-spin $e_g$ manifold near the Γ point, only one $\sigma^+$ circularly polarized transition is allowed among all the transitions, e.g., $\langle c_3|\hat{p}_+|v_1\rangle \neq 0$. Here $|v_1\rangle$ denotes the first valence state, $|c_3\rangle$ denotes the third conduction state and $\hat{p}_+ = \frac{-1}{\sqrt{2}}(\hat{p}_x + i\hat{p}_y)$ is the momentum operator in spherical basis with an angular momentum equal to 1; $\sigma^\pm$ denotes the circularly polarized light with the complex electric field amplitude along the direction of the spherical basis: $\mathbf{e}_\pm = \frac{\mp}{\sqrt{2}}(\mathbf{e}_x \pm i\mathbf{e}_y)$, where $\mathbf{e}_x$ ($\mathbf{e}_y$) is the unit vector along the $+x$ ($+y$) direction. This conclusion is further confirmed by our first-principles circularly polarized absorption shown in Fig. 3j. The $2s$-like exciton B$^-$, unlike A and B$^+$, is dominated by $\sigma^-$ circularly polarized transitions. We quantify the MCD of absorbance by calculating the contrast, $\eta = \frac{\text{Abs}(\sigma^+) - \text{Abs}(\sigma^-)}{\text{Abs}(\sigma^+) + \text{Abs}(\sigma^-)}$, where $\text{Abs}(\sigma^\pm)$ denotes the absorbance of $\sigma^+$ and $\sigma^-$ circularly polarized light, respectively. $\eta$ is dominated by $\sigma^+$ circularly polarized light below 1.8 eV (Fig. 3k). If we flip the magnetization direction, $\eta$ will also flip sign at all frequencies, which agrees with the measured MCD of photoluminescence signals[3].

In the following, we investigate the MO Kerr and Faraday effects of ferromagnetic monolayer CrI$_3$. Previous studies have shown that both SOC and the exchange splitting should be present to ensure non-zero MO effects in ferromagnets[18–22], and recent calculations within an independent-particle picture using DFT have been carried out for the MO responses of monolayer CrI$_3$[23]. The essence of a theoretical modeling of the MO effects lies in accurately accounting for the diagonal and off-diagonal frequency-dependent macroscopic dielectric functions, which are readily available from our $GW$-BSE calculations with electron−hole interaction included. We find that the

above-discussed giant excitonic effects in ferromagnetic monolayer CrI$_3$ strongly modify its MO responses, leading to significantly different behaviors going beyond those from a treatment considering only transitions between noninteracting Kohn−Sham orbitals[23]. Here we shall only consider the most physically relevant measurement for 2D ferromagnets, namely, polar MOKE (P-MOKE) and polar FE (P-FE), where both the sample magnetization and the wave vectors of light are along the normal of the surface. In accordance to typical, realistic experimental setup, we consider a device of ferromagnetic monolayer CrI$_3$ on top of a SiO$_2$/Si substrate (the thickness of SiO$_2$ layer is set to 285 nm, and Si is treated as semi-infinitely thick)[2], as shown in Fig. 4a. For insulating SiO$_2$ with a large bandgap (8.9 eV), we use its dielectric constant $\varepsilon_{\text{SiO}_2} = 3.9$[24]. For silicon, we perform first-principles $GW$ (at the $G_0W_0$ level) and $GW$-BSE calculations, and incorporate the frequency-dependence of the complex dielectric function $\varepsilon_{\text{Si}}(\omega)$ (see Supplementary Fig. 4). Assuming an incident linearly polarized light, we calculate the Kerr (Faraday) signals by analyzing the polarization plane of the reflection (transmission) light, which is in general elliptically polarized with a rotation angle $\theta_K(\theta_F)$ and an ellipticity $\chi_K$ ($\chi_F$) (see Supplementary Fig. 5). Here we adopt the sign convention that $\theta_K$ and $\theta_F$ are chosen to be positive if the rotation vector of the polarization plane is parallel to the magnetization vector, which is along the $+z$ direction.

We find that the MO signals are very sensitive to the thickness of SiO$_2$ and to the photon frequency. As shown in Fig. 4c, d and Supplementary Fig. 6, the thickness of SiO$_2$ layer will strongly affect the MO signals, due to the interference of reflection lights from multiple interfaces[2]. Such interference has been accounted for with our three-interface setup in Fig. 4a. To analyze the relation between MO signals and dielectric functions, we also consider a simpler two-interface setup. For a two-interface setup with semi-infinitely thick SiO$_2$ layer, the Kerr angle $\theta_K$ (Fig. 4d, solid blue curve) is related to Im[$\varepsilon_{xy}$] (Fig. 4b, dashed blue curve) and therefore resonant with the exciton excitation energies; the Kerr ellipticity $\chi_K$ (Fig. 4d, dashed red curve), on the other hand, is proportional to Re[$\varepsilon_{xy}$] (Fig. 4b, solid blue curve). For a two-interface model, $\theta_K$ is also found to be proportional to $n_0/(n_2^2 - n_0^2)$, where $n_0$ ($n_2$) is the refractive index for the upper (lower) semi-infinitely thick medium. Moreover, the $\theta_K$ and $\chi_K$ are connected through an approximate Kramers−Kronig relations, as expected from previous works[22,25]. Because of this, close attention should be paid in interpreting MOKE experiments on 2D ferromagnets, where the substrate configuration significantly changes the behavior of the MOKE signals. The existing experimental data of $\theta_K$, however, only have a few excitation frequencies of photons available, e.g., 5 ± 2 mrad at 1.96 eV for HeNe laser[2]. As shown in Fig. 4c, our simulations with a 285 nm SiO$_2$ layer in the three-interface setup achieve the same order of magnitude for $\theta_K$ around the MO resonance at ~1.85 eV, in good agreement with experiment. Based on the simulations, we also predict a sign change of $\theta_K$ around 1.5 eV. For photon energies higher than the quasiparticle bandgap, the plasmon resonance along with a vanishing $\varepsilon_{xx}$ will nullify our assumptions of continuous waves[25,26]. It is also possible to achieve an in-plane ferromagnetic structure with an external magnetic field[27,28]. However, due to the broken $C_3$ symmetry therein, we expect the system to have diminished values of MO signals (in the same polar configurations) but to remain having excitons with large binding energies, as confirmed by our first-principles calculations (see Supplementary Fig. 7).

**Effects of quantum confinement.** To further understand the effects of quantum confinement in 2D magnets, we compare the

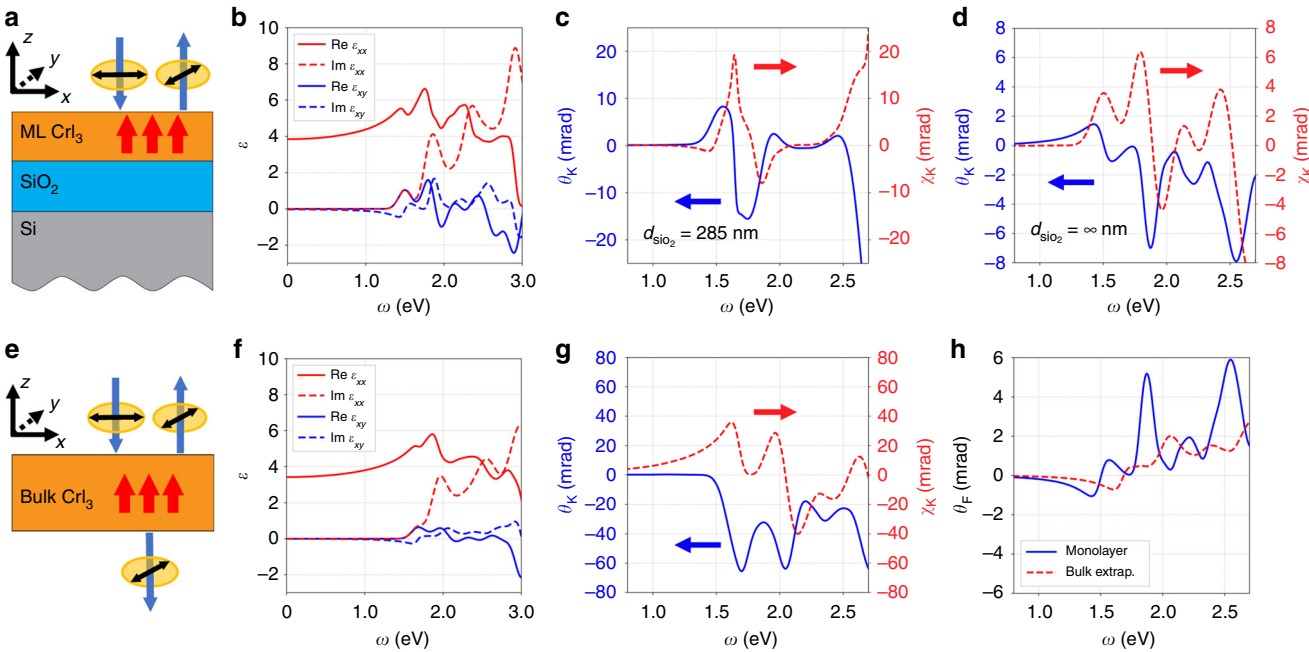

**Fig. 4** MO signals calculated from first-principles dielectric functions. **a** P-MOKE setup consisting of layers of vacuum, ferromagnetic monolayer $CrI_3$, $SiO_2$ film, and semi-infinitely thick Si. Red arrows denote the out-of-plane magnetization, which is pointing along the $+z$ direction. Blue arrows denote the propagation direction of light, and black double-headed arrows give the corresponding linear polarization direction. Each orange ellipse denotes a polarization plane of the electric field of light. **b** Calculated real part (solid lines) and imaginary part (dashed lines) of both the diagonal $\varepsilon_{xx}$ (red) and off-diagonal $\varepsilon_{xy}$ (blue) dielectric functions of ferromagnetic monolayer $CrI_3$, using a monolayer thickness $d = 6.6$ Å. An 80 meV energy broadening is applied. **c** Kerr angle $\theta_K$ (left, blue solid) and Kerr ellipticity $\chi_K$ (right, red dashed) for the P-MOKE setup with a 285 nm $SiO_2$ layer. **d** Kerr angle $\theta_K$ (left, blue solid) and Kerr ellipticity $\chi_K$ (right, red dashed) for the P-MOKE setup in (**a**) with semi-infinitely thick $SiO_2$ layer. **e** P-MOKE and P-FE setup of a suspended ferromagnetic bulk $CrI_3$ layer with the directions of light propagation and magnetization similar to (**a**). **f** Calculated real part (solid lines) and imaginary part (dashed lines) of both the diagonal $\varepsilon_{xx}$ (red) and off-diagonal $\varepsilon_{xy}$ (blue) dielectric functions of ferromagnetic bulk $CrI_3$, with an 80 meV energy broadening. **g** Kerr angle $\theta_K$ (left, blue solid) and Kerr ellipticity $\chi_K$ (right, red dashed) for the setup in (**e**) with infinitely thick ferromagnetic bulk $CrI_3$. **h** Comparison between Faraday angle $\theta_F$ of a suspended ferromagnetic monolayer $CrI_3$ and extrapolated bulk value down to the monolayer thickness (6.6 Å)

MO properties of ferromagnetic bulk and monolayer $CrI_3$. Interestingly, the calculated optical properties of bulk $CrI_3$ are also dominated by strongly bound excitons with optical absorption edge starting from 1.5 eV (in good agreement with experiment[3]), while the quasiparticle indirect bandgap is 1.89 eV and the direct bandgap at $\Gamma$ is 2.13 eV (see Supplementary Fig. 8). Within a one-interface model of semi-infinitely thick bulk $CrI_3$, $\theta_K$ reaches a magnitude of 60 mrad at the resonances at around 1.7 and 2.0 eV (Fig. 4g), proportional to $\mathrm{Re}[\varepsilon_{xy}]$ shown in Fig. 4f. To study the quantum confinement effect, we employ the P-FE setup shown in Fig. 4e, because P-FE in this setup is almost linear with respect to the ferromagnetic sample thickness and free from the substrate effects. Our calculated magnitude of the specific Faraday angle ($|\theta_F|$) of bulk $CrI_3$ is $(1.3 \pm 0.3) \times 10^3$ rad cm$^{-1}$ at the excitation frequency of 1.28 eV, in agreement with the experimental value of $1.9 \times 10^3$ rad cm$^{-1}$ at the same excitation frequency[29]. By extrapolating the bulk $\theta_F$ to the monolayer thickness[30], and comparing with that of suspended ferromagnetic monolayer $CrI_3$ as shown in Fig. 4h, we find that quantum confinement significantly enhances the MO response by a factor of 2.5 near 2.0 eV and introduces a redshift of 0.2 eV.

## Discussion

In summary, from our first-principles calculations, we discover that the optical and MO properties of ferromagnetic monolayer $CrI_3$ are dominated by strongly bound excitons of charge-transfer or Wannier characters. A systematic modeling framework for P-MOKE and P-FE experiments is also developed, where we have shown that the MO signals exhibit a sensitive dependence on

photon frequency and substrate configuration. These findings of the exciton physics in 2D magnets should shed light on design principles for future magneto-optical and optoelectronic devices, such as photo-spin-voltaic devices[31] and spin-injecting electroluminescence[32,33]. As a prototypical monolayer Ising magnetic insulator with a bandgap in an easily accessible optical range, ferromagnetic monolayer $CrI_3$ is also expected to be useful in high-speed and high-density flexible MO drives using van der Waals homostructures or heterostructures[27,34].

## Methods

**First-principles GW and GW-BSE calculations.** First-principles calculations of the electronic structure of ferromagnetic monolayer $CrI_3$ (as the mean-field starting point of the $G_0W_0$ and BSE studies) were performed at the DFT-LSDA level, as implemented in the Quantum ESPRESSO package[35], with parameters for the on-site Hubbard interaction $U = 1.5$ eV and Hund's exchange interaction $J = 0.5$ eV[8]. A slab model with a 16 Å vacuum thickness was adopted to avoid interactions between periodic images. We employed optimized norm-conserving Vanderbilt pseudopotentials including Cr 3s and 3p semicore states[36,37]. The Kohn−Sham orbitals were constructed with plane-wave energy cutoff of 80 Ry. Experimental structure was used in the calculations for both the bulk and monolayer $CrI_3$, with the lattice constants: $a = 6.867$ Å[30] (see Supplementary Table 3). SOC was fully incorporated in our calculations. The $GW$ (at $G_0W_0$ level) and $GW$-BSE calculations, for the quasiparticle and optical properties, respectively, were performed using the BerkeleyGW package[38]. The dielectric cutoff was set to 40 Ry. We adopted a $6 \times 6 \times 1$ grid with six subsampling points for calculating the dielectric function in ferromagnetic monolayer $CrI_3$[39]. An $18 \times 18 \times 1$ grid was then used for calculating the self-energy corrections. We treated the dynamical screening effect through the Hybertsen−Louie generalized plasmon-pole model[5], and the quasiparticle bandgap was converged to within 0.05 eV. The resulting quasiparticle band structure was interpolated with spinor Wannier functions, using the Wannier90 package[40]. Within our $GW$-BSE calculations, the exciton interaction kernel was interpolated from an $18 \times 18 \times 1$ grid to a $30 \times 30 \times 1$ grid using a linear interpolation scheme[9], and the transitions between 21 valence bands and 14 conduction

bands were considered in order to converge the calculation of the transverse dielectric functions from the $GW$-BSE results. The $GW$ (at $G_0W_0$ level) and $GW$-BSE calculations of ferromagnetic bulk $CrI_3$ used identical energy cutoffs and convergence thresholds as of monolayer $CrI_3$, and we adopted a $4 \times 4 \times 4$ grid for calculating the dielectric function and self-energy corrections in bulk $CrI_3$. The $GW$-BSE calculations of bulk $CrI_3$ employed a coarse grid of $6 \times 6 \times 6$ which was further interpolated to a fine grid of $10 \times 10 \times 10$. In this work, we obtained the calculated dielectric function of a ferromagnetic monolayer in a supercell slab model by using a thickness of a monolayer $CrI_3$ of $d = c_{bulk}/3 = 6.6$ Å (see Supplementary Figs. 9 and 10). Our calculations were performed for suspended $CrI_3$ in vacuum. Addition of an insulating substrate, such as fused silica or hexagonal boron nitride (hBN), introduces a small redshift of the exciton energies (estimated to be less than 0.1 eV for an hBN substrate, see Supplementary Fig. 11), while the strong excitonic effects still dominate the optical and MO responses. Effects of the on-site Hubbard potential on single-particle energies were systematically investigated to reveal the strong $p-d$ hybridization of the major-spin $e_g$ states (see Supplementary Fig. 12 and Supplementary Table 4).

**Group theory analysis**. We analyzed the symmetry of wave functions in ferromagnetic monolayer $CrI_3$. Ferromagnetic monolayer $CrI_3$ has point group symmetry $S_6 = C_3 \otimes C_i$. The irreducible representations labeled in Fig. 3i are for the double group $S_6^D$ due to the presence of strong spin−orbit coupling. The notation of the irreducible representations follows previous works[41,42].

## Data availability
The data that support the findings of this study are available from the corresponding author upon reasonable request.

## Code availability
The BerkeleyGW package is available from berkeleygw.org.

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

## Acknowledgements
The work was supported by the Theory Program at the Lawrence Berkeley National Lab (LBNL) through the Office of Basic Energy Sciences, U.S. Department of Energy under Contract No. DE-AC02-05CH11231, which provided the $GW$ & $GW$-BSE calculations and simulations and by the National Science Foundation under Grant No. DMR-1508412 and Grant No. EFMA-1542741, which provided for theoretical formulation and analysis of the MOKE simulations. Advanced codes were provided by the Center for Computational Study of Excited-State Phenomena in Energy Materials (C2SEPEM) at LBNL, which is funded by the U.S. Department of Energy, Office of Science, Basic Energy Sciences, Materials Sciences and Engineering Division under Contract No. DE-AC02-05CH11231, as part of the Computational Materials Sciences Program. Computational resources were provided by the DOE at Lawrence Berkeley National Laboratory's NERSC facility and the NSF through XSEDE resources at NICS.

## Author contributions
S.G.L. conceived the research direction and M.W. proposed the project. M.W. developed the full-spinor methods/codes, carried out computations and wrote the manuscript; M.W., Z.L. and T.C. analyzed the data; S.G.L. directed the research, proposed analyses and interpreted results. All authors discussed the results and edited this manuscript.

## Additional information

**Competing interests:** The authors declare no competing interests.

