## [Peer Review File · Nature Communications]

Reviewers' comments:

Reviewer #1 (Remarks to the Author):

In this work the authors calculate via the Bethe Salpeter equation on top of GW (performed on top of LDA+U using the de-screening procedure explained in S1) the excitonic spectrum in CrI₃.

Moreover they calculate the dichroic absorbance and the magneto optical kerr effect. The results are interesting and deserve publications in some form.

I have, however, some questions the authors should address as some parts of the manuscript are not completely clear.

1) The authors seem to suggest that it is important to do GW on top of LDA+U. However, looking at Fig. S1 it seems that LDA (U=0, J=0) already has the correct occupations (i.e. presence of a gap) and that different values of U lead to similar band structures at the LDA+U level. Surprisingly there are still differences at the GW level on top of LDA+U level (top of valence band at the M point).

I would expect that if GW is carried out self-consistently these differences should disappear. The authors then must have used the G0W0 approximation. This should be stated more clearly. Furthermore, it is to me unclear why the authors must use LDA+U as starting point of G0W0.

The absorbance has even larger differences so that it is difficult to see which spectrum should be the more reliable. My opinion is that the dependency of the spectra on U is the weakness of the paper. The authors should address more clearly this point in the manuscript.

2) By looking at Fig. S1 and comparing the LDA+U bands for (U,J)=(0,0), (U,J)=(0.5,0) and (U,J)=(2,0), it seems that for larger values of U the gap (at zone center) decreases instead of increasing.

Can the author rationalize a bit more this non-intuitive effect and explain it more in details? It would be nice also to have the equivalent of Fig. S3 for the U=0, J=0 case.

3) Concerning the crystal structure used in the calculation, is it the one optimized (cell? Internal coordinates?) with LDA, LDA+U, LDA+U+J I could not find the details of the structural optimization. It would be nice to have in the supplemental materials the details of the minimized structures (atomic positions, cell parameters, space group).

Reviewer #2 (Remarks to the Author):

The manuscript by Wu et al. reports on mechanisms of giant magneto-optical (MO) responses in two-dimensional (2D) ferromagnetic insulators, specifically CrI₃ by use of GW approximation with first principle calculations. The authors argue that the optical and MO responses of monolayer (ML) CrI₃ are dominated by strongly bound excitons, which have charge-transfer or Wannier-Mott excitonic nature. In addition to this, the authors found that the MO signals significantly depended on the photon energy and substrate configuration, that is bulk or SiO₂/Si. All these theoretical results are original and interesting works,

focusing the physical origin of giant MO effect in ML CrI₃, although the present manuscript is, in my opinion, not novel enough to be published in high impact journals, like Nature Communications.

This manuscript falls into recently noticeable field, 2D ferromagnetic insulators, which have been extensively investigated owing to their possible potential applications, such as spintronic devices, e.g., B. Huang et al. Nature Nanotech. Vol.13, 544-548 (2018). So theoretical studies on physical origin of giant MO effect in CrI₃ is timely and interesting issue. The main new idea of the present manuscript is that strongly bound excitons of charge transfer or Wannier-Mott character play a dominant role in the MO responses. Importantly, the binding energies from 0.3 eV for the bright exciton C to 1.7 eV for dark exciton D were obtained. These huge exciton binding energies imply the luminescence (or absorption) by these excitons may be possible to occur even at elevated temperature higher than room temperature. Similarly, the large exciton binding energies in 2D van der Waals semiconductors were obtained theoretically to be 0.1-0.2 eV for bulk MoTe₂ as found using GW-BSE *ab initio* calculations in Arora et al., Nature Comms. Vol.8, 639 (2017), and 0.3-0.4 eV for 1ML MoS₂ using tight-binding *ab initio* calculations with BSE by Ridolfi et al., in Phys. Rev. B. Vol.97, 205409 (2018), indicating unusually small exciton Bohr radius, typically on the order of 5 angstroms [Wu et al., Phys. Rev. B Vol.91, 075310 (2015)], but still of the Wannier-Mott type. In the present study, the exciton binding energies of 0.3-1.7 eV theoretically found in ML CrI₃ are really huge. Therefore, more careful discussion for the excitonic effects is required, as listed below.

(i) In general, exciton binding energy E_b is related to the Bohr radius a_B , that is the smaller E_b corresponding to the larger a_B [Wu et al., Phys. Rev. B Vol.91, 075310 (2015)]. Therefore, the larger E_b means smaller a_B . The bright exciton state A has the huge binding energy of ~ 1.1 eV, and therefore a_B would be really small. It may be interesting if the authors comment on this issue (for example Fig. 2g) in terms of 2D-exciton for CrI₃.

(ii) The exciton binding energy E_b (or the Bohr radius a_B) can be also related to the dielectric constant of CrI₃. For instance, $a_B \approx 13.6$ angstroms was obtained for $\text{Im}[\epsilon] \approx 12$ for MoS₂ [Wu et al., Phys. Rev. B Vol.91, 075310 (2015)]. It would be interesting if the authors introduce such the discussion.

(iii) In the 131st line, the authors noted the calculated circularly polarized absorption was consistent with the experimental photo-luminescence in Ref. [3]. Please explain which data in Ref. [3] is mentioning.

(iv) Jiang et al reported based on density functional theory calculations that applying the magnetic field along a-axis in CrI₃ causes a direct-to-indirect bandgap transition [Nano Lett. Vol.18, 3844-3849 (2012)]. Since the exciton effect would be eliminated in indirect bandgap state, it is interesting if the authors investigate or comment on the effect of in-plane magnetic polarization in Fig. 4.

In summary, I cannot recommend the publication of this manuscript in Nature Communications in its present form, but the current version should be strongly revised and improved before the paper could be reconsidered for the publication.

Reviewer #3 (Remarks to the Author):

In this manuscript the authors presented a detailed study of the exciton states in monolayer CrI₃ and the corresponding magneto-optic effects. Given the current interest in two-dimensional magnets, this result will certainly be welcomed by the community. The same group has been churning out papers on 2D excitons and is certainly a leading group in this field. However, I do

think this paper can be significantly improved by a more detailed analysis of their result and shedding some new light on the physics involved.

The technical achievement here is the including of the spin-orbit coupling from the outset of the calculation, which, as far as I am aware of, few groups can do properly. The authors claimed good agreement between their calculation and experiment. In particular, they calculated the absorbance spectrum and identified three peaks, A, B, and C, which are assigned to interatomic p-d transitions. However, this is in contradiction with the ligand field theory analysis in Ref. 3 and Ref. 15. By looking at the exciton wave function in Fig 2, it is clear that the A exciton is still highly localized within the unit cell and B is also somewhat confined, so a molecular view of the optical transition should still be valid to some degree. It'll be interesting to make the connection between the ligand field theory and the band theory in the context of optical transitions, and particular when the connection breaks down. I strongly recommend the author added such an analysis to (i) make the paper have a broader appeal to the broader community since the ligand field theory is widely used in chemistry and (ii) shed some new light on the two complementary theories, i.e., ligand field theory vs. band theory. For example, the B- exciton wave function has a hole at the Gamma point (Fig. 3), does this mean it can no longer be described within the molecular picture? And what was missing in the ligand field theory that leads to the contradiction?

In addition, does the substrate modify the exciton energy? Is the result in Fig. 2 for a monolayer in vacuum or with substrate?

I should mention that Fig 4 is also very interesting. The sign change of the Kerr angle as a function of frequency should be an easily testable result.

In summary, I think the result presented in this paper is very interesting, but the paper could be improved by more analysis.

Response to Reviewer #1's Comments

Reviewer #1's comment #1:

“The authors seem to suggest that it is important to do GW on top of LDA+U. However, looking at Fig. S1 it seems that LDA ($U=0, J=0$) already has the correct occupations (i.e. presence of a gap) and that different values of U lead to similar band structures at the LDA+U level. Surprisingly there are still differences at the GW level on top of LDA+U level (top of valence band at the M point). I would expect that if GW is carried out self-consistently these differences should disappear. The authors then must have used the G0W0 approximation. This should be stated more clearly. Furthermore, it is to me unclear why the authors must use LDA+U as starting point of G0W0. The absorbance has even larger differences so that it is difficult to see which spectrum should be the more reliable. My opinion is that the dependency of the spectra on U is the weakness of the paper. The authors should address more clearly this point in the manuscript.”

Our response:

We thank the reviewer for these insightful comments. The choice of U , according to our calculations, does have some influence on the quasiparticle band energies and wavefunction hybridization, but would not alter the giant excitonic effects and exciton-dominant nature of the magneto-optical (MO) signals. For a good starting point of the G_0W_0 calculation, we carefully checked a range of U and J , and pinned their values down to $U=1.5$ eV & $J=0.5$ eV for optimal agreement with the experimental optical spectra although a slight variation in their values does not change any of the physical conclusions.

The effect of U and J on the quasiparticle band energies and wavefunction hybridization could be seen both in the hybridization between the Cr d and I p orbitals and the energy splitting between the Cr major-spin t_{2g} and minor-spin e_g manifolds of bands. With $U=J=0$ (Fig. S1a), these two sets of bands touch and the band states are mixed around Γ point. By increasing U , we see a gradual separation between these two sets of states. A good LSDA+ U starting point should then keep similar band orders

(within the localized d states) and energy separation compared to those of the subsequent G_0W_0 calculations. (In this sense, we crudely achieve a level of consistency within a quasiparticle self-consistent framework.) We calculated the G_0W_0 self-energy and bandstructure (with $U = 1.5$ eV & $J = 0.5$ eV) on a dense $18 \times 18 \times 1$ grid as shown in Fig. 1b. The similarity of energy splitting and dispersion of these bands (dominated by spin-polarized d states) between LSDA+U and $G_0W_0@LSDA+U$ justifies our usage of LSDA+U with $U = 1.5$ eV & $J = 0.5$ eV as the starting point for the subsequent G_0W_0 calculations. The appropriateness of this starting point can also be confirmed by the GW-BSE calculations on the photo absorption (Fig. 2c & Fig. S2). With $U = 1.5$ eV & $J = 0.5$ eV, the calculated photo absorption with the GW-BSE method has good agreement with experiments in both peak positions and amplitudes. Deviations between results from $U=J=0$ (Fig. S2a) and $U = 1.5$ eV & $J = 0.5$ eV (Fig. 2c) in fact support our usage of LSDA+U (with U and J in a reasonable range) as the starting point for G_0W_0 , as is also evidenced in the PDOS plots in Fig. S3. The observation that there are still differences around the top of valence band at the M point can be explained by the character change of the VBM states (less contributions from d states with increasing U). Further analysis on the MO signals and good agreement with experiments (Fig. 4) reassure us the reliability of LSDA+U with $U = 1.5$ eV & $J = 0.5$ eV as a good starting point.

The fact that LSDA ($U=J=0$) alone has the correct occupation is expected from the crystal field and the exchange interaction that already exist in LSDA: the octahedral environment leads to the well-known partitioning of t_{2g} and e_g ligand states, while the exchange interaction leads to further separation between these states with major-spin and minor-spin polarizations.

As noted by the Reviewer, we adopted the G_0W_0 approximation instead of the self-consistent GW approach. The exact self-energy Σ could in principle be obtained by self-consistently solving a set of integrodifferential equations^{R1}. However, it is known that the omission of the vertex correction (that would introduce higher order interactions) with every iteration in such a procedure makes a self-consistent solution inconsistent and worsens the spectral properties of the Green's function^{R2, R3}. In addition, self-consistent GW calculations including off-diagonal element effects are currently formidably expensive, especially for CrI_3 systems of which a large number of bands appear near the Fermi level. This is why the G_0W_0 approximation has been adopted, which is believed to yield accurate quasiparticle bandstructure compared to the full-frequency self-consistent GW_0 calculations^{R4} Future

works involving self-consistent GW calculations are of course valuable and necessary to get rid of the dependency on the DFT starting point, but they are beyond the scope of current effort in the understanding of magneto-optical phenomena.

In response to the comment and to clarify the generality of the physics here, we have added a sentence in the second paragraph about our computational method:

“
In practical calculations, the G_0W_0 approximation⁵ is adopted in our study where the GW self-energy is treated as a first-order correction, and the single-particle Green's function G and the screened Coulomb interaction W are calculated using eigenvalues and eigenfunctions from density-functional theory (DFT).
”

We have also replaced the term “GW” with “ G_0W_0 ” everywhere in the manuscript to avoid potential misunderstanding.

Reviewer #1's comment #2:

“By looking at Fig. S1 and comparing the LDA+U bands for $(U,J)=(0,0)$, $(U,J)=(0.5,0)$ and $(U,J)=(2,0)$, it seems that for larger values of U the gap (at zone center) decreases instead of increasing. Can the author rationalize a bit more this non-intuitive effect and explain it more in details? It would be nice also to have the equivalent of Fig. S3 for the $U=0, J=0$ case.”

Our response:

We thank the reviewer for raising this question. This non-intuitive effect of reduced bandgap (by a small amount) with increasing U is rooted in the strong p - d hybridization and deviation of the e_g states from a simple atomic picture. In CrI_3 , the e_g ligand states are derived from σ bonds between Cr d orbitals and I p orbitals. Because the Hubbard U correction relies on the wavefunction overlap integral between the hybridized Bloch states and the localized occupied atomic orbitals, such σ bonds leads to a small overlap integral and therefore a small energy correction. For the VBM state at the Γ point, the wavefunction is dominated by I p states, resulting in little change of band energy upon $+U$. In addition, we find small but noticeable overlap between the major-spin e_g states and the occupied major-spin

diagonal orbitals (used in calculating the Hubbard potential), which reduces the CBM energy by a small amount (proportional to U) according to the *Janak theorem* (described in SI, Sec. K).

In response, we have revised the first paragraph to clarify the observed p - d hybridization:

“

Although the major-spin e_g states are delocalized due to strong p - d hybridization, the magnetic moment is approximately $3\mu_B$ at each Cr site, in accordance with the atomic picture from the first Hund's rule⁴.

”

To address this comment in more details, we have summarized the above-mentioned effect and added a new section (Sec. K) in SI,

“

K. Effect of U on single-particle energies

There is an interesting behavior of LSDA+ U band energies with increasing U . As shown in the PDOS plots in **Fig. S10**, in contrast to the common impression of effects of + U , the bandgap and the first conduction peak (both of which contain substantial p orbital character) slightly decreases upon increasing U . Other Cr d states, on the other hand, show expected behaviors by going away from the Fermi level with increasing U .

Figure S10. PDOS plots of ferromagnetic monolayer CrI_3 with different U values from (a) $U = 0$ eV to (e) $U = 2.0$ eV. J is set to be zero in all cases for this illustrative investigation. The DOS is projected into contributions from Cr d (red) and I p (blue) orbitals. The green dots show the trends of band energies with increasing U . The single-particle energies have been aligned with the vacuum level in all the plots. The dashed black line indicates the VBM energy.

The effects of U could be understood in terms of the *Janak theorem*: the eigenvalue is the derivative of the total energy with respect to the occupation of a state i , $\epsilon_i = \frac{\partial E_{tot}}{\partial n_i}$, where i also includes the spin index^{S11}. In the case of LSDA+ U , the total energy is given by $E_{tot} = E_{LSDA} + E_U - E_{dc}$ ⁸, and therefore the Bloch state eigenvalue may be expressed as,

$$\epsilon_{nk} = \frac{\partial E_{LSDA}}{\partial n_{nk}} + \frac{\partial (E_U - E_{dc})}{\partial n_{nk}}. \quad (\text{S17})$$

To simplify our following discussions, we now assume $J=0$. The energy correction from + U can be estimated as,

$$\Delta\epsilon_{nk}^{+U} = \sum_{at, m\sigma} U \left(\frac{1}{2} - n_{m\sigma} \right) |\langle m\sigma | \psi_{nk} \rangle|^2, \quad (\text{S18})$$

where $n_{m\sigma}$ is the occupation number for the *diagonalized* and spin-polarized local basis $|m\sigma\rangle$, and the Bloch states $|\psi_{nk}\rangle$ are assumed to be fixed before and after +U. In this way, the eigenvalue correction should incorporate an extra factor to quantify how much the projection of the Bloch state is onto the localized basis. This is in contrast to the atomic limit, $\Delta\epsilon_i^{+U} = U(\frac{1}{2} - n_i)$, which indicates that the occupied and unoccupied states at the LSDA level are further split by U. But there is strong *p-d* hybridization in the CrI₃ systems, especially for e_g states which form σ bonds between Cr *d* orbitals and I *p* orbitals. In this way, the major-spin e_g states become delocalized, which means the overlap between the first set of conduction Bloch states and the localized atomic orbital $|m\sigma\rangle$ used in LSDA+U is small. To be explicit, $\Delta\epsilon_{nk}^{+U}$ should be smaller for the e_g states than for the t_{2g} states, which has also been verified in our calculations. Moreover, the strong *p-d* hybridization makes the local occupation $n_{m\sigma}$ deviate from the atomic limit: that is, 3 major-spin *d* orbitals are completely occupied, while all the remaining *d* orbitals are empty. Here we used U = 2.0 eV in a case study. As listed in Table S2, the spin density has a more uniform distribution than that in the atomic limit.

U=2 eV & J=0 eV	Minor spin					Major spin					$\Delta\epsilon_{+U}$ (eV)	
$n_{m\sigma}$	0.118	0.121	0.129	0.397	0.400	0.708	0.713	0.993	0.993	0.993	Janak	LSDA+U
$ \langle n_{m\sigma} v_1 @ \Gamma \rangle ^2$	1.1E-5	2.4E-5	8.3E-5	1.1E-5	1.3E-4	1.6E-9	3.0E-5	6.1E-3	1.6E-5	1.2E-2	-0.04	-0.01
$ \langle n_{m\sigma} c_1 @ \Gamma \rangle ^2$	7.4E-5	2.0E-6	6.6E-4	1.5E-6	1.2E-4	3.1E-7	2.2E-4	6.7E-2	1.6E-4	8.6E-2	-0.30	-0.16
$ \langle n_{m\sigma} c_5 @ \Gamma \rangle ^2$	1.6E-1	1.6E-1	5.4E-2	6.5E-3	2.4E-2	6.4E-4	1.0E-5	1.0E-5	1.2E-6	1.2E-6	0.58	0.63
$ \langle n_{m\sigma} c_{14} @ \Gamma \rangle ^2$	1.9E-2	7.0E-3	3.6E-2	3.2E-6	2.0E-1	5.9E-6	2.4E-6	2.1E-5	9.1E-7	2.1E-5	0.17	0.27

Table S2. Occupation of diagonal basis on each Cr atom of a ferromagnetic monolayer CrI₃ from LSDA+U calculations with U = 2.0 eV & J = 0 eV. The local spin-resolved occupation matrix $n_{ij}^{\sigma\sigma'}$ is diagonalized with eigenvalues $n_{m\sigma}$ as shown in the second row. The projections of the 1st valence state and the 1st, 5th, 14th conduction states at the Γ point to these localized diagonal orbitals are calculated in the following rows. The Janak analysis and the LSDA+U results on the change in the band energy due to +U are listed in the last two columns.

Also note that the local spin-resolved occupation is completely determined from the occupied Bloch states, while the overlap between unoccupied Bloch states and occupied diagonal local basis ($n_{m\sigma} > 0.5$) is not necessarily zero! Indeed, our numerical results agree with our intuition that the e_g states will be

more delocalized and have smaller overlap with localized diagonal orbitals. Also, the empty major-spin e_g states have small but noticeable overlap between the occupied major-spin diagonal orbitals, giving rise to the negative shift of CBM energy with increasing U , as confirmed from both our Janak analysis and DFT calculations (last two columns in Table S2). Moreover, the p -dominant VBM state (v_I) at the Γ point has little overlap with those d -like localized diagonal orbitals, leading to negligible energy shift upon $+U$. For these reasons, the bandgap slightly decreases with increasing U .

”

As requested by Reviewer #1, we have added a PDOS plot for the $U=J=0$ case in Fig. S3:

“

Figure S3. Partial density of states of a ferromagnetic monolayer CrI₃ for (a) $U = 0$ eV, $J = 0$ eV and (b) $U = 1.5$ eV, $J = 0.5$ eV. The density of states is decomposed into contributions from Cr major-spin $3d$ orbitals (red solid curve), Cr minor-spin $3d$ orbitals (blue dashed curve), I major-spin $5p$ orbitals (green dotted curve) and I minor-spin $5p$ orbitals (black solid curve). The energy of the VBM is set to zero.

”

Reviewer #1’s comment #3:

“Concerning the crystal structure used in the calculation, is it the one optimized (cell ? Internal coordinates ?) with LDA, LDA+U, LDA+U+J I could not find the details of the structural optimization. It would be nice to have in the supplemental materials the details of the minimized structures (atomic positions, cell parameters, space group).”

Our response:

We appreciate Reviewer #1’s comment about structure relaxation. As described in **Methods**, we used the experimental structure for both ferromagnetic bulk and monolayer CrI₃ in our calculations. We have also performed structure relaxation of the ferromagnetic bulk and monolayer structure, and we found good agreement with the experimental structure and little change of atomic positions from bulk to monolayer in both cases. In response, we have added a section (Sec. L) in SI to address Reviewer #1’ concern:

“

L. Crystal structure and structure relaxation

We used the experimental structure for both ferromagnetic bulk and monolayer CrI₃. Bulk CrI₃ belongs to the space group $R\bar{3}(148)$ ³⁰. Both bulk and monolayer belong to the point group S_6 . In fact, we have checked the validity of the employed pseudopotentials with first-principles structure relaxation. During the relaxation, we used a kinetic energy cutoff of 120 Ry. The van der Waals (vdW) interaction is included in two ways: the rVV10 nonlocal density functional^{S12} and the semiempirical Grimme’s DFT-D3 method^{S13}, both of which have been implemented in the Quantum ESPRESSO package³⁵. The structures were fully relaxed until the force on each atom was less than 0.02 eV/Å. The spin-orbit coupling effects were not considered in the relaxation. We found that there was little deviation between the relaxed bulk structure and the experimental bulk structure. In addition, the lattice constants and internal coordinates barely changed from bulk to monolayer structures, indicating much stronger intralayer bondings than interlayer bondings. For these reasons, we just used the experimental structure for both bulk and monolayer calculations. The detailed structure parameters are listed in Table S3.

	Parameters (Å)	a	c	Cr-I distance	Interlayer distance	Intralayer thickness
Bulk	Exp. ³⁰	6.87	19.81	2.73	3.47	3.13
	rVV10	6.99	19.70	2.78	3.38	3.19

	DFT-D3	6.96	20.07	2.76	3.53	3.16
Monolayer	rVV10	6.98	N/A	2.78	N/A	3.21
	DFT-D3	6.96	N/A	2.76	N/A	3.18

Table S3. Structure parameters for ferromagnetic bulk and monolayer CrI₃. The lattice constants a and c refer to the conventional hexagonal cell.

The Quantum ESPRESSO structure input for bulk CrI₃ (with fractional coordinates) is listed below:

CELL_PARAMETERS angstrom

```
3.9648952386328364 0.0000000000000000 6.602333333333333
-1.9824476193164182 3.4337000000000000 6.602333333333333
-1.9824476193164182 -3.4337000000000000 6.602333333333333
```

ATOMIC_POSITIONS crystal

```
Cr 0.666330330 0.666330330 0.666330330
Cr 0.333669670 0.333669670 0.333669670
I 0.729082000 0.430059000 0.077768000
I 0.270918000 0.569941000 0.922232000
I 0.077768000 0.729082000 0.430059000
I 0.922232000 0.270918000 0.569941000
I 0.430059000 0.077768000 0.729082000
I 0.569941000 0.922232000 0.270918000
```

The Quantum ESPRESSO structure input for monolayer CrI₃ (with fractional coordinates) is listed below:

CELL_PARAMETERS angstrom

```
6.8670001030 0.0000000000 0.0000000000
-3.4335000515 5.9469965370 0.0000000000
0.0000000000 0.0000000000 18.0000000000
```

ATOMIC_POSITIONS crystal

```
Cr 0.333333333 0.666666667 0.000377778
Cr 0.666666667 0.333333333 0.999622226
```

I 0.349896669 0.998803318 0.913106084
I 0.001196661 0.351093352 0.913106084
I 0.648906648 0.650103331 0.913106084
I 0.650103331 0.001196663 0.086893886
I 0.998803318 0.648906648 0.086893886
I 0.351093352 0.349896699 0.086893886
”

Response to Reviewer #2's Comments

We thank Reviewer #2's highly positive comment that “*All these theoretical results are original and interesting works, focusing the physical origin of giant MO effect in ML CrI₃.*” However, we strongly disagree with Reviewer #2's comment that this manuscript lacks novelty. As Reviewer #2 pointed out: “*theoretical studies on physical origin of giant MO effect in CrI₃ is timely and interesting issue.*”, our work is the first to study excitonic effects in magneto-optics from first principles. Previous works either focus on the MO effects in magnetic metals in which excitonic effects are weak, or simply ignore the excitonic effects in magnetic insulators. The giant MO effect in monolayer ferromagnets, a real puzzle to the community, had not been fully understood prior to our work. The novelty of our work lies in the indepth physical understanding of and the discovery of dominant excitonic effects in MO effects in ferromagnetic monolayer and bulk CrI₃, by applying a newly developed GW & GW-BSE methods that incorporate spin-orbit coupling, onsite Coulomb interaction, and most importantly first-principles many-electron treatment of electron-hole interactions. This has, to the best of our knowledge, never been done before. In the following, we provide our responses to the three points raised by Reviewer #2. We have also revised our manuscript and SI accordingly.

Reviewer #2's comment #1:

“In general, exciton binding energy R is related to the Bohr radius a_B , that is the smaller R corresponding to the larger a_B [Wu et al., Phys. Rev. B Vol.91, 075310 (2015)]. Therefore, the larger R means smaller a_B . The bright exciton state A has the huge binding energy of ~ 1.1 eV, and therefore a_B would be really small. It may be interesting if the authors comment on this issue (for example Fig. 2g) in terms of 2D-exciton for CrI₃.”

Response:

We thank Reviewer #2 for his/her comments on the relation between exciton binding energy and the exciton radius. Indeed, this comment agrees very well with our calculations. Within a hydrogenic model, it is a well-known result that $R \propto \frac{1}{a_B}$ ^{R5, R6}, and therefore a large R means a small a_B . Tight-binding and first-principles GW-BSE calculations in other 2D materials, such as transition metal dichalcogenides and black phosphorus, have also confirmed the above conclusion that a large binding energy indicates a small exciton radius^{R7-R9}. To have a numerical understanding of this conclusion, we evaluated the relation between the exciton binding energy and the exciton radius for exciton states in Fig. 2 and Fig. 3 (see Table S4). Our numerical results have confirmed the inverse proportionality relation between R and a_B while deviations might come from the multi-band effects and reduced long-range screening beyond the 2D hydrogenic model.

In response, we have revised the manuscript by adding a sentence to explain this intuition about Fig. 2g,

“

This agrees with the intuition that a smaller exciton binding energy is related to a larger exciton radius¹¹,¹². Numerical calculations of the exciton radius further corroborate this conclusion (see SI).

”

We have added a new reference in the manuscript as Ref. 12:

“

12. Wu, F. et al. Exciton band structure of monolayer MoS₂. *Phys. Rev. B* **91**, 075310 (2015).

“

In addition, we have added a new section (Sec. M) in SI to present the numerical evidences about the above analysis:

“

M. Numerical evaluation of exciton radius

We used the exciton wavefunctions shown in Fig. 2 and Fig. 3 to evaluate the arithmetic mean radius $\langle|r|\rangle$ and the root mean square radius $\sqrt{\langle r^2 \rangle}$ as shown in Table S4. Our first-principles results agree with the intuition that a large binding energy indicates a small exciton radius, as inspired by a 2D hydrogenic model.

Exciton states	A	B ⁺	B ⁻	C
Ω_S (eV)	1.50	1.82	1.92	2.31
E_b (eV)	1.09	0.77	0.67	0.28
$\langle r \rangle$ (Å)	2.33	3.55	5.36	6.99
$\sqrt{\langle r^2 \rangle}$ (Å)	3.06	4.37	6.70	7.93

Table S4. The arithmetic mean radius $\langle|r|\rangle$ and root mean square radius $\sqrt{\langle r^2 \rangle}$ of selected bright exciton states. The excitation energy Ω_S and binding energy E_b are also included for reference.

”

Reviewer #2’s comment #2:

“The exciton binding energy R (or the Bohr radius a_B) can be also related to the dielectric constant of CrI3. For instance, $a_B \approx 13.6$ angstroms was obtained for $\text{Im}[\epsilon] \approx 12$ for MoS2 [Wu et al., Phys. Rev. B Vol.91, 075310 (2015)]. It would be interesting if the authors introduce such the discussion.”

Response:

We thank the reviewer for raising this interesting question. In a simplified 2D hydrogenic model with constant screening independent of electron-hole separate, it is well known from quantum mechanics that the binding energy and orbital radius are related to the dielectric constant through the relations: $R \propto \frac{1}{\epsilon^2}$ and $a_B \propto \epsilon^{R5}$. However, more recent studies have discovered that the dielectric function for a 2D materials is not a constant, but strongly depends on the distance between the two charged quasiparticles ^{R6, R10, R11}. This conclusion has also been confirmed by previous works on black phosphorus and transition metal dichalcogenides ^{R9, R12-R14}. So, we feel that an oversimplified discussion would do more harm than help in elucidating the real physics involved here.

Reviewer #2’s comment #3:

“In the 131st line, the authors noted the calculated circularly polarized absorption was consistent with the experimental photo-luminescence in Ref. [3]. Please explain which data in Ref. [3] is mentioning.”

Response:

We thank the reviewer for pointing out this point. In this sentence, we refer to Fig. 1c and Fig. 2d in Ref. 3.

To avoid potential misunderstanding, we have revised this sentence as:

“

Note that our calculated circularly polarized absorption is consistent with measured circularly polarized photoluminescence when the magnetization is polarized along the $-z$ direction³.

”

Reviewer #2's comment #4:

“Jiang et al reported based on density functional theory calculations that applying the magnetic field along a -axis in CrI₃ causes a direct-to-indirect bandgap transition [Nano Lett. Vol.18, 3844-3849 (2012)]. Since the exciton effect would be eliminated in indirect bandgap state, it is interesting if the authors investigate or comment on the effect of in-plane magnetic polarization in Fig. 4.”

Response:

We thank Reviewer #2 for this interesting question; however, we disagree with the referee that “*the exciton effect would be eliminated in indirect bandgap state*”. In fact, indirect bandgap materials, such as hexagonal boron nitride (hBN)^{R15} and anatase TiO₂^{R16}, are known to exhibit strongly bound exciton just as direct bandgap materials do. In SI Sec. N, we present results from LSDA+U and G₀W₀ calculations reproducing the magnet band structure effect with an in-plane magnetic polarization, as discussed by Jiang et al^{R17}. Moreover, our BSE calculations have demonstrated similarly strong excitonic effects as in the case of out-of-plane magnetic polarization. From a symmetry analysis, we have also predicted and then numerically verified the diminished polar MO responses in this case.

In response, we have added a sentence in the manuscript to comment on the effects of in-plane magnetic polarization:

“

It is also possible to achieve an in-plane ferromagnetic structure with an external magnetic field^{27, 28}. However, due to the broken C_3 symmetry therein, we expect the system to have diminished values of

MO signals (in the same polar configuration) but to remain having excitons with large binding energies, as confirmed by our first-principles calculations (see SI).

”

We have also added three new references in the manuscript as Ref. 14, Ref. 23 and Ref. 28:

“

14. Wirtz, L. et al. Excitons in boron nitride nanotubes: dimensionality effects. *Phys. Rev. Lett.* **96**, 126104 (2006).

23. Gudelli, V. K. and Guo, G.-Y. Magnetism and magneto-optical effects in bulk and few-layer CrI₃: a theoretical GGA+U study. arXiv:1902.03944 (2019).

28. Jiang, P. et al. Spin direction-controlled electronic band structure in two-dimensional ferromagnetic CrI₃. *Nano Lett.* **18**, 3844 (2018).

”

Moreover, we have added a new section (Sec. N) in SI:

“

N. In-plane ferromagnetic monolayer CrI₃

Our LSDA+U and G₀W₀ calculations have reproduced the reported magneto band structure effect where relevant bands are degenerate at the Γ point for the case of an in-plane ferromagnetic monolayer CrI₃²⁸. We have obtained an indirect bandgap of 2.64 eV at the G₀W₀ level in the in-plane polarized structure, which is only slightly smaller than the direct bandgap at the Γ point (2.69 eV). The rotated magnetization has a strong impact on the polar MO signals, which can be understood from the symmetry. The dielectric function tensor ϵ is an axial tensor, which is in analogue to a dyad of two vectors. The broken C_3 rotational symmetry in an in-plane polarized structure leads to diminished amplitudes of ϵ_{xy} and ϵ_{yx} , because the x - and y - directions are no longer correlated. We therefore expect small polar MO signals in an in-plane polarized structure. Our first-principles GW-BSE calculations confirmed the above analysis as shown in Fig. S11f regarding to the polar MO Faraday effect. Moreover, we have found that there are still strong excitonic effects in the in-plane case. In fact, we could still identify the three excitonic peaks (A, B and C) below the quasi-particle bandgap (Fig. S11d), with the exciton A having a binding energy of 1.29 eV, even larger than that in the out-of-plane case. This is probably

because the high band degeneracy in the in-plane case increases the joint density-of-states around the Γ point, further enhancing the excitonic effects.

Figure S11. (a) Crystal structure and magnetization direction (red arrows) of an in-plane ferromagnetic monolayer CrI_3 . (b) G_0W_0 (red solid) and LSDA+U (blue dashed) band structure of the in-plane ferromagnetic monolayer CrI_3 , where a Hubbard onsite potential with $U = 1.5$ eV & $J = 0.5$ eV is

adopted. (c) Exciton levels of in-plane ferromagnetic monolayer CrI₃ from GW-BSE calculations. Bright excitons are colored in red while dark ones in blue. The continuum starts from 2.69 eV. (d) Linearly polarized absorption spectrum at normal incidence with (GW-BSE, red solid) and without (GW-RPA, blue dashed) electron-hole interaction. An 80 meV energy broadening is adopted in (d) and the following plots in (e) and (f). (e) Calculated real part (solid lines) and imaginary part (dashed lines) of ϵ_{xx} (red), ϵ_{yy} (green), ϵ_{xy} (blue) and ϵ_{yx} (yellow) dielectric functions of in-plane ferromagnetic monolayer CrI₃. (f) Comparison between Faraday angle θ_F of an out-of-plane ferromagnetic monolayer CrI₃ and an in-plane ferromagnetic monolayer CrI₃ with the same P-FE configuration as in Fig. 4e.

”

Response to Reviewer #3's Comments

We thank Reviewer #3's very positive comments: *“Given the current interest in two-dimensional magnets, this result will certainly be welcomed by the community.”* and *“The technical achievement here is the including of the spin-orbit coupling from the outset of the calculation, which, as far as I am aware of, few groups can do properly.”* In the following, we provide point-to-point responses to the comments from Reviewer #3. Following his/her suggestions, we have revised our manuscript and SI accordingly.

Reviewer #3's comment #1:

“By looking at the exciton wave function in Fig 2, it is clear that the A exciton is still highly localized within the unit cell and B is also somewhat confined, so a molecular view of the optical transition should still be valid to some degree. It'll be interesting to make the connection between the ligand field theory and the band theory in the context of optical transitions, and particular when the connection breaks down.

For example, the B- exciton wave function has a hole at the Gamma point (Fig. 3), does this mean it can no longer be described within the molecular picture? And what was missing in the ligand field theory that leads to the contradiction?”

Response:

We thank Reviewer #3 for this insightful question. We agree with Reviewer #3 that ligand field theory is widely used in chemistry to interpret the electronic structure of transition-metal and rare-earth compounds^{R18}. We also agree with the reviewer that ligand field theory could provide a qualitative description in terms of orbital characters and suppressed oscillator strength of the dark D exciton and the slightly bright A exciton, because these two tightly bound exciton states are mainly composed of intra-atomic *d-d* transitions, with a uniform distribution of exciton amplitude $A_{c\mathbf{v}\mathbf{k}}^S$ in *k*-space (Fig. 3a&b). However, ligand field theory has certain limitations in interpreting the optical transitions of CrI₃. First, it fails to give a quantitative evaluation of the finite oscillator strength for the A exciton, as well as higher-lying bound exciton states and continuum resonant exciton states, which are needed for the MO responses. Second, bound exciton states covering several unit cells (such as B and C states in this work, as shown in Fig. 2h-k) are beyond ligand field theory, because multiple band-pair transitions are involved and the optical transitions only happen in a small fraction of the Brillouin zone (Fig. 3c&d). First-principles GW-BSE calculations, on the other hand, could capture these effects simultaneously and produce quantitative agreement with experiments. Therefore, we believe that our first-principles GW-BSE method is more suitable for this problem, because it provides an effective two-particle equation that captures the detailed description of the exciton wave functions and provide a quantitative description of all the relevant exciton states, covering the whole range of exciton radius from atomic scale to the many-unit-cell scale. It should also be emphasized that our first-principles treatment of nonlocal dielectric screening effects, strong spin-orbit coupling effects, as well as strong Coulomb interactions between electrons and holes, is crucial to the full understanding of the underlying optical and MO physics of CrI₃.

In response, we have revised the manuscript by adding the following sentences:

“

In fact, ligand field theory provides a qualitative understanding of the lowest-lying D and A exciton states of which the optical transitions mainly occur among the localized Cr *d* orbitals^{3, 17}. However, ligand field theory is insufficient to evaluate the oscillator strength of the excitons quantitatively. In addition, the coexistence of Frenkel and Wannier excitons in our system poses significant challenges to ligand field theory, while this excitonic physics can be fully captured by the first-principles GW-BSE method.

”

To emphasize the importance of excitonic effects on MO signals, we have added a sentence:

“

We show that the above-mentioned giant excitonic effects in ferromagnetic monolayer CrI₃ strongly modify its MO responses, leading to significantly different behaviors going beyond those from a treatment considering only transitions between non-interacting Kohn-Sham orbitals²³.

”

Reviewer #3's comment #2:

“In addition, does the substrate modify the exciton energy? Is the result in Fig. 2 for a monolayer in vacuum or with substrate?”

Response:

We thank Reviewer #3 for these interesting questions. The results in Fig. 2 are for a suspended monolayer in vacuum. As for the substrate effects, our conclusion is that the influence of an insulating substrate, such as hexagonal boron nitride (hBN) or fused silica, will only cause a small *redshift* of the exciton excitation energies in ferromagnetic monolayer CrI₃. As for metallic substrates, such as monolayer and few-layer graphene, we expect a larger reduction of the exciton binding energies due to the metallic screening^{R14}. Since the relevant photo absorption and MOKE experiments were done with insulating fused silica^{R19}, sapphire^{R20} or hBN^{R21} substrates, we focus on the case of an insulating substrate in the following. Previous experimental measurements and theoretical studies have shown that the exciton energies stay nearly constant for various substrate configurations with a small shift, which is attributed to a nearly exact cancellation between the change in quasiparticle band gap and exciton binding energy^{R14, R20, R22-R24}. For example, the A exciton energy in monolayer transition metal dichalcogenides has been measured to be largely unaffected by the surrounding medium^{R8, R14, R25}. The fact that ferromagnetic bulk and few-layer CrI₃ has a similar optical gap around 1.5 eV is also reminiscent of this nearly perfect cancellation effect^{R20}. This picture is particularly true for insulating 2D materials where the electronic states are mainly confined within each atomic layer, and the substrate only couple weakly to the material through the dielectric screening.

In a case study, we performed GW (at G₀W₀ level) and GW-BSE calculations of a monolayer CrI₃ on top of a monolayer hBN with an interlayer distance of 3.42 Å from structure relaxation. Since hBN has a

dielectric constant very similar to that of SiO₂ (both around 4), hBN and fused silica substrates of the same thickness should have similar screening effects on the exciton energies, as confirmed in previous work on transition metal dichalcogenides^{R14}. It is expected that the substrate-induced exciton excitation energy redshift, albeit small, will quickly saturate with increasing thickness and additional layers of substrate would introduce negligible deviations^{R14}. According to our first-principles calculations, we found a redshift of less than 0.1 eV for the whole optical excitation spectrum (Fig. S12), while the quasiparticle bandgap is 2.41 eV with the hBN substrate compared with 2.59 eV in the suspended case. In particular, the A exciton energy is 1.46 eV with the hBN substrate while it is 1.50 eV without substrate. The oscillator strength for the exciton states are also largely unaffected by the addition of the hBN substrate (Fig. S12c).

In response, we have added a sentence in **Methods** to clarify our computational approach and discussed the effects of an insulating substrate on the exciton energies:

“

Our calculations were performed with suspended CrI₃ in vacuum. Addition of an insulating substrate, such as fused silica or hexagonal boron nitride (hBN), introduces a small *redshift* of the exciton excitation energies (estimated to be less than 0.1 eV for a hBN substrate in SI), while the strong excitonic effects still dominate the optical and MO responses.

”

We have also added a new section (Sec. O) in SI:

“

O. Effect of a hexagonal BN (hBN) substrate on exciton energies

To study the effects of an insulating substrate, we performed GW-BSE calculations of a ferromagnetic monolayer CrI₃ on top of a monolayer hBN with an interlayer distance of 3.42 Å. The interlayer distance was determined with the van der Waals interaction through the DFT-D3 method^{S13}. It is expected that the substrate-induced exciton excitation energy redshift, albeit small, will quickly saturate with increasing thickness and additional layers of substrate will introduce negligible deviations^{S14-S16}. Since hBN has a dielectric constant very similar to that of SiO₂ (both around 4), hBN and fused silica substrates of the same thickness should have similar screening effects on the exciton energies, as confirmed in previous work on transition metal dichalcogenides^{S14}.

Figure S12. (a) Schematic of a ferromagnetic monolayer CrI₃ on top of a monolayer hexagonal BN substrate, with an interlayer distance of 3.42 Å. A Hubbard onsite potential with $U = 1.5$ eV and $J = 0.5$ eV is adopted. (b) Exciton levels of ferromagnetic monolayer CrI₃ with a monolayer hexagonal BN substrate from GW-BSE calculations. Bright excitons are colored in red while dark ones in blue. The continuum starts from 2.41 eV. (c) Linearly polarized absorption spectrum at normal incidence with (GW-BSE, red solid) and without (GW-RPA, blue dashed) electron-hole interaction. An 80 meV energy broadening is adopted.

”

References

- R1. Hedin, L. *Phys. Rev.* **139**, A796 (1965).
- R2. F. Aryasetiawan and O. Gunnarsson, *Rep. Prog. Phys.* **61**, 237 (1998).
- R3. P. Rinke, A. Qteish, J. Neugebauer, and M. Scheffler, *Phys. Status Solidi B* **245**, 929 (2008).
- R4. Lischner, J. et al, *Phys. Rev. B* **90**, 115130 (2014).
- R5. Yang, X. L. et al. *Phys. Rev. A* **43**, 1186 (1991).
- R6. Cohen, M. L. and Louie, S. G. *Fundamentals of Condensed Matter Physics*, Cambridge University Press, 2016.
- R7. Wu, F. et al. *Phys. Rev. B* **91**, 075310 (2015).
- R8. Ye, Z. et al. *Nature* **513**, 214 (2014).
- R9. Qiu, D. Y. et al. *Nano Lett.* **17**, 4706 (2017).
- R10. Keldysh, L. V. *JETP Lett.* **29**, 658 (1978).

- R11. Qiu, D. Y. et al. *Phys. Rev. B* **93**, 235435 (2016).
- R12. Ugeda, M. M. et al. *Nat. Mater.* **13**, 1091 (2014).
- R13. Chernikov, A. et al. *Phys. Rev. Lett.* **113**, 076802 (2014).
- R14. Raja, A. et al. *Nat. Comm.* **8**, 15251 (2017).
- R15. Cassabois, G. et al. *Nat. Photon.* **10**, 262 (2016).
- R16. Baldini, E. *Nat. Comm.* **8**: 13 (2017).
- R17. Jiang, P. et al. *Nano Lett.* **18**, 3844 (2018).
- R18. Ballhausen, C. J. Introduction to Ligand Field Theory, McGraw Hill, 1962.
- R19. Huang, B. et al. *Nature* **546**, 270 (2017).
- R20. Seyler, K. L. et al. *Nat. Phys.* **14**, 277 (2018).
- R21. Zhong, D. et al. *Sci. Adv.* **3**, e1603113 (2017).
- R22. Chernikov, A. et al. *Phys. Rev. Lett.* **115**, 126802 (2015).
- R23. Arenal, R. et al. *Phys. Rev. Lett.* **95**, 127601 (2005)
- R24. Ross, J. S. et al. *Nat. Comm.* **4**:1474 (2013).
- R25. Lin, Y. et al. *Nano Lett.* **14**, 5569 (2014).

REVIEWERS' COMMENTS:

Reviewer #1 (Remarks to the Author):

The authors correctly replied to all questions by the referees. My opinion is that this paper is a bit borderline for Nature Communications, as there is not a very strong message. However, given the fact that there are new technical achievements and that the material is very hot in this moment, I finally think that the paper should be published in nature communications.

Some minor corrections:

when they explain why DFT+U closes the gap instead of opening it, there is no definition of what is the state $|n_m\sigma\rangle$ (although there is definition of $n_m\sigma$). Maybe the authors should explain it a bit better.

Reviewer #2 (Remarks to the Author):

I have read through both the revised manuscript and the response letter by Wu et al. In my opinion, the authors made effective changes by taking the suggestions of all the referees into account carefully and improved the discussion and clarity of the interpretation. Publication in the current version is recommended.

Reviewer #3 (Remarks to the Author):

The authors have answered all my questions. I have no more comments and recommend publication in Nature Comm.

Reviewer #1

Comments:

“when they explain why DFT+U closes the gap instead of opening it, there is no definition of what is the state $|n_m\sigma\rangle$ (although there is definition of $n_m\sigma$). Maybe the authors should explain it a bit better.”

Responses:

We thank Reviewer #1 for pointing out this issue.

In response, we have added the following paragraph in Supplementary Information to address this comment:

“

To get $n_{m\sigma}$ and $|m\sigma\rangle$, we first use atomic wave functions (five spin-up d orbitals and five spin-down d orbitals for each Cr atom) as projectors to build the local orbital-resolved spin density $n_{ll'}^\sigma$ by summing over all the contributions from occupied Bloch states, with the quantization axis chosen along the z -axis. Here $l, l' = 1, \dots, 5$ labels the d orbitals and $\sigma = \uparrow, \downarrow$ labels the spin polarization. We then diagonalize this matrix of spin density and get the eigenvalues n_m^σ and eigenvectors $|m\sigma\rangle$ as a linear combination of the five d orbitals with spin polarization σ .

”